# TRPA1 Expression in Synovial Sarcoma May Support Neural Origin

**DOI:** 10.3390/biom10101446

**Published:** 2020-10-15

**Authors:** Francesco De Logu, Filippo Ugolini, Chiara Caporalini, Annarita Palomba, Sara Simi, Francesca Portelli, Domenico Andrea Campanacci, Giovanni Beltrami, Daniela Massi, Romina Nassini

**Affiliations:** 1Section of Clinical Pharmacology and Oncology, Department of Health Sciences, University of Florence, 50139 Florence, Italy; Francesco.delogu@unifi.it (F.D.L.); romina.nassini@unifi.it (R.N.); 2Section of Pathological Anatomy, Department of Health Sciences, University of Florence, 50139 Florence, Italy; filippo.ugolini@unifi.it (F.U.); annarita.palomba@aouc.unifi.it (A.P.); sara.simi@unifi.it (S.S.); francesca.portelli@unifi.it (F.P.); 3Pathology Unit, Meyer Children’s Hospital, 50139 Florence, Italy; chiara.caporalini@meyer.it; 4Orthopedics and Traumatology Section, Department of Health Sciences, University of Florence, 50139 Florence, Italy; domenicoandrea.campanacci@unifi.it (D.A.C.); giovanni.beltrami@unifi.it (G.B.)

**Keywords:** TRPA1, synovial sarcoma, neural stem cells

## Abstract

Synovial sarcoma (SS) is a malignant mesenchymal soft tissue neoplasm. Despite its name, the cells of origin are not synovial cells, but rather neural, myogenic, or multipotent mesenchymal stem cells have been proposed as possible cells originators. Unlike other sarcomas, an unusual presentation of long-term pain at the tumor site has been documented, but the exact mechanisms have not been fully clarified yet. The transient receptor potential ankyrin 1 (TRPA1) is a nonselective cation channel mainly expressed in primary sensory neurons, where it functions as a pain sensor. TRPA1 have also been described in multiple non-excitable cells, including those derived from neural crest stem cells such as glial cells and, in particular, Schwann cell oligodendrocytes and astrocytes. We evaluated TRPA1 expression in SS. We selected a cohort of 41 SSs, and by immunohistochemistry, we studied TRPA1 expression. TRPA1 was found in 92.6% of cases. Triple TRPA1/pS100/SOX10 and TRPA1/SLUG/SNAIL staining strongly supports a neural origin of SS. TRPA1 positivity was also observed in a subset of cases negative with pS100, SOX10 and/or SLUG/SNAIL, and these divergent phenotypes may reflect a process of tumor plasticity and dedifferentiation of neural-derived SSs. Given the functional diversity of TRPA1 and its expression in neuronal and non-neuronal multipotent neural crest stem cells, it remains to be determined whether TRPA1 expression in SSs neoplastic cells plays a role in the molecular mechanism associated with premonitory pain symptoms and tumor progression.

## 1. Introduction

The transient receptor potential ankyrin 1 (TRPA1) is a calcium-permeable nonselective cation channel belonging to the large family of TRP channels. TRPs are expressed in almost every cell type, where they mediate pleiotropic functions in different physiological and pathological conditions. Apart from its prominent expression in sensory neurons [1], where it functions as a sensor of tissue damage, TRPA1 has been detected in multiple non-neuronal cells, including cell lineages derived from neural crest stem cells, such as glial cells, and in particular Schwann cells [2], oligodendrocytes [3] astrocytes [4], and other cells, including melanocytes [5] and chondrocytes [6].

Synovial sarcoma (SS) is a malignant mesenchymal neoplasm with variable epithelial differentiation that accounts for approximately 7 to 10% of all soft tissue sarcomas [7,8]. According to the World Health Organization classification of bone and soft tissue tumors, SS represents a distinct morphological, clinical, and genetically defined tumor entity, with a propensity to occur in adolescents and young adults, which can arise in almost any site [9]. SS are divided into two major histologic subtypes, biphasic and monophasic [10,11], on the basis of cellular components. Genetically, SSs harbor a highly specific t(X;18) (p11.2;q11.2) translocation producing a SYT-SSX fusion gene [12], which is detected in over 90% of SS, regardless of histologic subtype [9].

Its origin is widely debated and currently unknown, although a neural, myogenic, or a multipotent mesenchymal stem cell origin have all been considered [12,13,14]. Genome-wide analysis [15] and mRNA and protein analyses of neural tissue genes in primary SS and cell lines [12] reported the expression of several genes typical of neural crest-derived cells, capable of differentiating in multiple directions and migrating into many tissue types. Moreover, the varied and ubiquitous distribution of SS may support the hypothesis of a stem cell origin, suggesting neural crest stem cells as possible candidate precursors of SS [12].

Although the diagnostic gold standard for SS has long been the use of molecular and cytogenetic tools, such as fluorescence in situ hybridization (FISH), or reverse transcription-polymerase chain reaction (RT-PCR), immunohistochemical testing remains more widely available to most pathology practices [10].

Immunohistochemically, more than 90% of SSs demonstrate focal expression of epithelial markers, cytokeratins (CK7, CK8, CK18, CK19), and epithelial membrane antigen (EMA). A vast majority of SSs are positive for CD99 and the transducer-like enhancer of split 1 (TLE1) [10]. Focal expression of S100 protein (pS100) can be detected in 8 to 63% of cases [16,17,18,19], and pS100 expression may be also be taken to support neural crest differentiation. The Sry-Related HMG-Box Gene 10 (SOX10) is known to be a transcription factor implicated in neural crest differentiation into Schwann cells [20,21]. Limited immunohistochemical data [22] showed that SOX10 is expressed in less than 10% of SSs. Whether SOX10-positive cells in intraneural SS represent entrapped Schwann cells, synovial sarcoma cells, or both, remains to be determined. 

Given the broad expression of TRPA1 in different cells from neural origin and the debated postulated cell of origin of SSs, here we evaluate the expression of TRPA1 protein in a series of SSs, to support the hypothesis of neural-stem cells as a source for SS differentiation.

## 2. Materials and Methods

### 2.1. Case Series

The study cohort included formalin-fixed paraffin-embedded (FFPE) tissues from 41 representative synovial sarcoma (34 primary SS, 2 recurrent SS and 5 metastatic SS). FFPE samples were retrospectively retrieved from the archive of the Division of Pathology, Department of Health Sciences, University of Florence, Italy. Patient data, including age and sex, were collected. The use of FFPE sections of human samples was approved by the Local Ethics Committee (#11979_bio, 16 July 2019), according to the Helsinki Declaration, and informed consent was obtained.

### 2.2. Tissue Samples

FFPE tissue sections, 3 μm in thickness, were stained with hematoxylin and eosin and centrally reviewed to assess pathology tissue quality control. SS diagnosis was confirmed by immunohistochemistry, and in doubtful cases, FISH analyses detected the specific translocation t(X;18) (SS18-SSX1/2,). (LSI FUS (16p11) dual color, break apart rearrangement probe; Abbot Molecular, IL, USA).

### 2.3. Immunohistochemistry

Representative 3-μm thick FFPE tissue sections of SS were selected for immunohistochemical and immunofluorescence analysis. Sample processing was performed with automated immunostainer (Ventana Discovery ULTRA, Ventana Medical Systems, Tucson, AZ, USA).

The sections were deparaffinized in EZ prep (950–102; Ventana Medical Systems, Tucson, AZ, USA), and antigen retrieval was achieved by incubation with cell-conditioning solution 1 (950–124; Ventana Medical Systems, Tucson, AZ, USA), tris ethylenediaminetetraacetic acid-based buffer (pH 8.2) or with cell-conditioning solution 2 (760–107; Ventana Medical Systems, Tucson, AZ, USA) citrate-based buffer (pH 6.5). Sections were then incubated with the following primary antibodies: anti-pS100 (#2666, RRID:AB_2335936, mouse monoclonal, clone 4C4.9 ready to use, Ventana Medical Systems, Tucson, AZ, USA), anti-pS100 (#2133, RRID:AB_2335920, rabbit polyclonal ready to use, Ventana Medical Systems, Tucson, AZ, USA), anti-SOX10 (#760–4968 rabbit monoclonal, clone SP267, ready to use, Ventana Medical Systems, Tucson, AZ, USA) anti-SOX10 (#BSB2581, rabbit monoclonal, clone EP268, ready to use, Bio SB, Santa Barbara, CA, USA), anti-TRPA1 (#ACC-037, RRID:AB_2040232, rabbit polyclonal, 1:100, Alomone Labs Ltd., Jerusalem, Israel) and anti-TRPA1 (#ab58844, RRID:AB_945957, rabbit polyclonal, 1:500, Abcam, Cambridge, UK) and anti-SNAIL+SLUG (#ab180714, RRID: AB 2728773, rabbit polyclonal, 1:500, Abcam Cambridge, UK). 

For immunohistochemistry, the signal was developed with the UltraMap diaminobenzidine anti-mouse or anti-rabbit detection kit (Ventana Medical Systems, Tucson, AZ, USA). Negative controls were performed by substituting the primary antibodies with normal serum for S100, SOX-10 and SNAIL+SLUG and pre-adsorption with immunizing peptide (overnight 4 °C) for TRPA1 antibodies. Sections were counterstained with hematoxylin. Stained tissue sections were digitally scanned at ×400 magnification with Aperio AT2 platform (Leica Biosystems, Wetzlar, Germany).

Tissue staining was assessed by evaluating the percentage of marked area compared to the total surface of the tissue, and by assigning an intensity value ranging from 0 (negative), 1+ (weak), 2+ (strong) to 3+ (very strong). In addition, TRPA1 staining was evaluated as the percentage of marked area compared to the total surface of the tissue and by a semi-quantitative H-score analysis in primary/metastatic and monophasic/biphasic SSs. For the H-score calculation, the percentage of TRPA1 positive cells (0% to 100%) was multiplied by the dominant intensity pattern of staining (0, no staining; 1, weak; 2, strong; 3, very strong); therefore, the overall score ranging from 0 to 300 was calculated. In this system, <1% positive cells were considered a negative result. 

## 3. Immunofluorescence

Sample processing was performed with automated immunostainer (Ventana Discovery ULTRA, Ventana Medical Systems, Tucson, AZ, USA). The sections were deparaffinized in EZ prep (950–102; Ventana Medical Systems, Tucson, AZ, USA), and antigen retrieval was achieved by incubation with cell-conditioning solution 1 (950–124; Ventana Medical Systems, Tucson, AZ, USA), tris ethylenediaminetetraacetic acid-based buffer (pH 8.2), for 36 min at 95 °C. Sections were blocked with blocking buffer (BB) (PBS, pH 7.4, 2.5% bovine serum albumin BSA, 10% normal goat serum, NGS) for 32 min at 37 °C, then incubated with a solution containing the following primary antibodies: anti-pS100 (mouse monoclonal, clone 4C4.9 ready to use, Ventana Medical Systems, Tucson, AZ, USA) and anti-TRPA1 (rabbit polyclonal, diluted 1:100, Alomone Labs Ltd., Jerusalem, Israel) diluted in antibody diluent (Roche Diagnostics, Mannheim, Germany). After another cycle of BB for 16 min at 37 °C, sections were incubated with fluorescent secondary antibodies: polyclonal Alexa Fluor 594 and polyclonal Alexa Fluor 647 (1:500, Invitrogen, Milan, Italy) (1 h 32 min, RT). Sections were coverslipped using a water-based mounting medium with 4′6′-diamidino-2-phenylindole (DAPI) (Abcam, Cambridge, UK). Fluorescence images were obtained using an AxioImager 2 microscope (Carl Zeiss, Oberkochen, Germany). Imaris software (Oxford Instruments, version 8.1.2, Zurich, Switzerland) was used for 3D reconstruction of the structured illumination Z-stacks. The colocalization of pS100 and TRPA1 was assessed by counting positive cells expressing both antigens at a magnification of 20× using an AxioImager 2 microscope (Carl Zeiss). Data are reported as percentage of S100^+^ and TRPA1^+^ cells of total TRPA1^+^ cells.

## 4. Statistical Analysis

The data are shown as mean ± standard deviation (SD). Statistical analysis was performed using the unpaired two-tailed Student’s t-test for comparisons between two groups (GraphPad Prism version 5.00, San Diego, CA, USA).

## 5. Results

The SS patients included 22 males and 19 females. The mean age was 42.6 years (range 15–78). Of the 41 analyzed samples, 34 cases were primary tumors, 2 were recurrent tumors, and 5 were metastatic lesions. The sites of involvement of primary tumors were upper extremities (*n* = 24), lower extremities (*n* = 7), buttocks (*n* = 2) and paravertebral region (*n* = 1). Recurrent tumors were localized in the right knee and left hand. The lung was the site of involvement of metastatic lesions. Tumors ranged in size from 2 to 10 cm. Histologically, of the 34 primary tumors, 27 (79%) were monophasic and 7 (20%) biphasic. The two recurrent tumors were both monophasic. Of the five metastatic cases, three (60%) were monophasic and two (40%) biphasic. All SSs analyzed contained mostly tumor cells (94%), normal cells constituted only a small part of the entire cell population (6%).

All 41 tissues were stained with pS100, SOX10 and TRPA1 antibodies. Two different antibodies from different sources and clones for each marker were used in order to check for any false-positives immune-staining. Results showed that 13/41 (31.7%) cases were positive to pS100 and 6/41 (14.6%) to SOX10. TRPA1 staining was detected in neoplastic cells in 38/41 (92.6%) of SSs examined. TRPA1 staining was identified mainly at the cytoplasmic level, although the proportion of cells that expressed protein was variable across the tumor sections analyzed, ranging from 5% to 90% (Figure 1A). TRPA1 negative staining was observed in 3/41 (7.3%), week/moderate staining TRPA1 protein staining was observed in 21/41 (51.2%) and strong/very strong staining was documented in 17/41 (41.5%) cases. Nuclear TRPA1 staining was reported in only 2/41 specimens.

There was no significant difference in TRPA1 expression and the nature of the tumor (primary vs recurrent/metastatic) or histotype (monophasic vs biphasic), either in the percentage of the marked area or in the H-Score analysis (Figure 1B,C). Data obtained with pS100, SOX10 and TRPA1 antibodies from a different source confirmed the previous staining intensity, thus excluding the presence of false-positives (data not shown).

TRPA1 expression has been previously reported in pS100^+^ and SOX10^+^ Schwann cells [2]. Data showed that 6/41 (14.6%) SSs positive with pS100 and SOX10 were also positive to TRPA1 staining. In addition, 6/41 (14.6%) SSs resulted in being positive to pS100 and TRPA1, but negative to SOX10, and 26/41 (63.4%) were positive to TRPA1, but negative to both pS100 and SOX10 (Figure 2). Among SSs negative with TRPA1 staining, a small percentage (1/41, 2.5%) resulted in being positive with pS100 staining, and 2/41 (4.9%) cases were negative to both pS100 and SOX10 (Figure 2). In addition, the expression of TRPA1 in pS100^+^ cells, was confirmed by a double staining (Figure 3). The analysis of the co-expression of pS100 and TRPA1 showed that 48% (±12%) of pS100^+^ cells also express the TRPA1 channel.

To further confirm a neural origin for SSs expressing TRPA1, different markers (SNAIL+SLUG), usually expressed by the stem cells of the neural crest, were tested. Among the SSs, 35/41 (85%) were positive with SNAIL+SLUG. In addition, data showed that 32/41 (78%) SSs positive with SNAIL+SLUG were also positive with TRPA1 staining; 6/41 (15%) SSs resulted in being positive with TRPA1, but negative with SNAIL+SLUG, and 3/41 (7%) were negative with TRPA1 and positive with SNAIL+SLUG (Figure 4).

## 6. Discussion

The present study is the first report investigating the presence of TRPA1 in a cohort of SS human tissue samples. Our data showed an overall staining (TRPA1 positivity in 92.6% of the examined SSs) for TRPA1 protein in the tested cases. TRPA1 staining was observed in neoplastic cells at the cytoplasmic level, with a high expression in approximately 41% of specimens.

TRPA1 is a member of the large TRP family of calcium ion channels, widely expressed in mammalian tissues, which functions as a calcium permeable non-selective cation channel in many different cell processes, ranging from sensory to homeostatic functions [23]. TRPA1 is broadly expressed in neuronal and non-neuronal cells derived from neural-stem cells. The most clearly defined physiological function of TRPA1 is in sensory neuronal cells, where it plays a crucial role in pain perception [23]. At these locations, TRPA1 works as a sensor of exogenous and endogenous chemical stimulants, such as allyl isothiocyanate, acrolein, and a wide range of oxidant products, including hydrogen peroxide, and byproducts of oxidative stress, such as 4-hydroxynonenal [24,25,26,27] produced during tissue injury. Besides its main expression in a subset of primary sensory neurons being a pain sensor, the presence and a functional role of TRPA1 in non-neuronal cells derived from neural-stem cells have been progressively shown. Recently, the presence of a functional TRPA1 protein in Schwann cells and its role in orchestrating neuroinflammation and ensuing neuropathic pain have been described [2]. TRPA1 has also been identified in oligodendrocyte, the analog of Schwann cells in the central nervous system, with possible detrimental roles in ischemia and neurodegeneration [3]. TRPA1 contributes to the calcium mediated neuronal hyperactivity induced by Aβ oligomers in early phases of Alzheimer’s disease [28]. Further observations have described the expression and a functional role of this channel in non-neuronal neural-stem cells. TRPA1 activation leads to an increase in intracellular calcium and a rapid increase in cellular melanin content in melanocytes [5], as well as the activation of the apoptosis pathway in chondrocytes, which represents a central feature in the progression of osteoarthritis [6]. Given its role in the pathophysiology of nearly all organ systems, TRPA1 represents an attractive target for the treatment of related diseases [29]. Thus, our study of TRPA1 protein expression in SSs may gain new insights on the biology of this tumor. SS is an aggressive malignant neoplasm accounting for 7–10% of soft-tissue sarcomas in adolescents and young adults that occurs in almost any anatomical site [9,30]. To date, few studies have reported TRP expression in sarcomas and, in particular, in SSs [31], although in recent years, the role of TRP channels in the progression and metastasis of different types of tumors has been gradually clarified [32,33]. The first study analyzing TRP channels in sarcomas reported a marked increase of the expression of the TRP melastatin (TRPM8) subfamily in patients with osteosarcoma at a higher clinical stage compared to those with a lower clinical stage and the absence of metastasis [34]. Thus, a higher TRPM8 expression seems to be related with an adverse prognosis [34]. To the best of our knowledge, only one in vitro study has reported an aberrant expression of members of the TRP canonical (TRPC4/C1) subfamily in human synovial sarcoma cell lines, with a detrimental effect upon activation [35].

However, considerable evidence suggests an increasing role of TRP channels as biomarkers for diagnosis and/or prognosis in different tumors, although it has not yet been established whether changes in channel expressions are drivers, or if they are only required to sustain the transformed phenotype [32,33]. For instance, among different TRP channels, the expression levels of members of the TRPC, TRPM, and vanilloid (TRPV) subfamilies have been correlated with the development and/or progression of certain epithelial cancers [36,37,38,39]. TRPM1 gene expression has been inversely correlated with the aggressiveness of melanoma malignant cells, thus suggesting its role as a tumor suppressor gene [36]. Another member of the TRPM subfamily, TRPM7, has been associated with a worse outcome in metastatic breast cancer [40]. Moreover, TRPV6 is strongly expressed in advanced prostate cancer, with little or no expression in healthy and benign prostate tissues [41], and high TRPC6 expression has been documented in esophageal squamous cell carcinoma [42].

Recently, TRPA1 has also been reported to mediate a non-canonical oxidative stress defense program in cancer cells and the upregulation of anti-apoptotic pathways to promote cancer cell survival [43]. Based on this evidence, we speculated that the presence of a high level of TRPA1 protein in SSs may suggest that the channel activation in SS neoplastic cells affects adaptive mechanisms involving the promotion of calcium mediated mechanisms able to induce a survival defense program in cancer cells and the ensuing tumor progression. In our cohort, no statistically significant differences in TRPA1 expression were demonstrated in primary vs. recurrent/metastatic lesions, as well as in the different histotypes (monophasic vs biphasic) SSs, but the limited sample size does not allow us to draw definitive conclusions. A larger cohort of SSs is required to better understand the overall expression and a possible significance of TRPA1 in SS progression.

SS is currently classified as a tumor of uncertain histogenesis characterized by a balanced reciprocal translocation t(X;18) and, usually, at least focal evidence of epithelial differentiation [9]. Until now, the cellular origin of SS remains to be resolved. Growing evidence suggests that some malignant tumors arise from tissue stem cells [44,45,46,47]. Among these, neural crest stem cells have been advocated as a possible candidate for the precursor of SSs [12]. This hypothesis is derived from evidence reporting that SSs mimic the morphological, immunophenotypical and clinical pattern of the MPNST, the precursor cells of which are neural crest-derived cells [15]. Furthermore, it has been extensively reported that the overexpression of genes associated with stem cell identity raises the possibility that these genes may contribute to stem cell-like SS phenotypes and aggressive tumor behavior [48,49,50,51]. Several studies [12,13,14,15] have reported the expression of a number of genes of neural crest-derived cells in primary SS and cell lines. In addition, the mechanism of deregulation of cell differentiation, driven by SYT-SSX fusion protein, stem cell transformation, and cancer development, has been widely hypothesized [13,14]. Recent findings have reported [13] that SYT-SSX fusion gene silencing induced the differentiation of SS cells into multiple mesenchymal lineages, supporting the hypothesis that SS arises in multipotent stem cells.

Among genes associated with neural crest stem cells, the nerve growth factor receptor (p75NTR) gene was expressed in a subset of SSs [12]. The SOX10 transcription factor, a putative marker of the differentiation of neural crest stem cells, may have diagnostic utility in this differential identification of SS origin [21], although it occurs in less than 10% of SSs. Furthermore, data comparing genetically confirmed synovial sarcoma with malignant peripheral nerve sheath tumor with respect to SOX10 expression are limited. S100 protein identification also possesses limited diagnostic utility, as higher grade MPNST are often negative and only up to a third of SSs are positive [19].

In our study, 41 cases of SSs have been examined. Consistent with previous data [52], we have reported that 31.7% of SSs tested expressed pS100, while SOX10 was observed in a smaller percentage (14.6%) of the SS specimens [24]. A subset of SSs showed a positive staining to more specific neural markers, such as SNAIL and SLUG. The TRPA1/pS100/SOX10 and TRPA1/SNAL/SLUG positivity strongly supports a neural origin in a subset of SS.

However, TRPA1 also resulted in being expressed in a subset of cases negative with pS100, SOX10 and/or SNAIL and SLUG, and we may speculate that these divergent phenotypes could reflect a process of tumor plasticity and dedifferentiation of neural-derived SSs.

Another important issue to be considered concerns the role of TRPA1 in promoting and maintaining a painful condition. In addition to its prominent role in primary sensory neurons as a detector of painful stimuli, a fundamental role for Schwann cell TRPA1 has been reported. In a murine model of neuropathic pain, TRPA1 activation in Schwann cells generates a spatially constrained gradient of oxidative stress, which maintains macrophage infiltration to the site of injury, and sends paracrine signals to activate TRPA1 of unsheathed nociceptors to sustain pain [2].

Interestingly, patients suffering from SS report pain symptoms arising directly from the lesion and refer to pain due to pressure [53,54]. Such pain, which is also seen in other sarcomas, is probably due to the expansion, stretching and destruction of surrounding tissues, and hemorrhage and necrosis within the tumor [55,56]. However, it has been reported that patients with SS are twelve times more likely to have pain at the site of the tumor prior to the development of swelling, compared to patients with other sarcomas affecting the extremities and trunk [55]. The mechanism for this premonitory pain has not been fully explained. These early pain symptoms occur when the tumors are small, thus pressure on the surrounding tissues or necrosis within the tumor are unlikely to be major causes of pain. Although the release of cytokines, prostaglandins and other inflammatory mediators has been supposed [57,58], the mechanism for this premonitory pain needs further investigation.

Following its activation in non-neuronal cells, TRPA1 also releases proinflammatory cytokines, including interleukin-8 [59]. In this perspective, due to its ability to release oxidant molecules and proinflammatory mediators, TRPA1 activation in SSs neoplastic cells could take part in the maintenance of an inflammatory status in the site of lesion, which, targeting sensory nerves surrounding the lesion, maintains pain. Additional studies on the downstream pathway related to TRPA1 activation in SS cell lines and the possibility of detecting TRPA1 in SSs at a pretumor stage may support major evidence on a pivotal role of this channel in SS-associated pain.

In summary, evidence on TRPA1 protein expression in a subset of SSs, also positive to pS100, SOX10 and SNAIL and SLUG, sustains previous studies on neural crest stem cells as possible histogenetic precursors. The novel finding of TRPA1 expression in SSs may open a discussion on its role as a player in the genesis and development of this aggressive tumor and, given the uncertainty of the mechanism, to the premonitoring of peculiar pain symptoms.

## Figures and Tables

**Figure 1 biomolecules-10-01446-f001:**
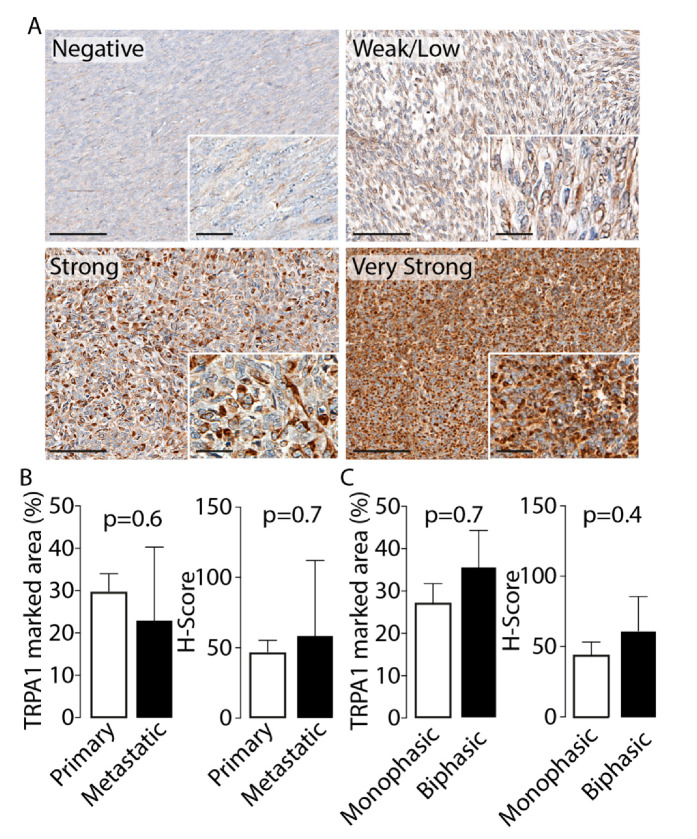
(**A**). Representative photomicrographs of transient receptor potential ankyrin 1 (TRPA1) staining intensity (negative, weak/low, strong and very strong) in synovial sarcoma (SS) samples. Cumulative data of TRPA1 marked area (%) and H-Score in (**B**) primary and metastatic and (**C**) monophasic and biphasic SS samples (**C**). Data are reported as mean ± SD, Student’s t-test. Magnification ×200, inset ×400 (Scale bars: 100 µm, inset 20 µm).

**Figure 2 biomolecules-10-01446-f002:**
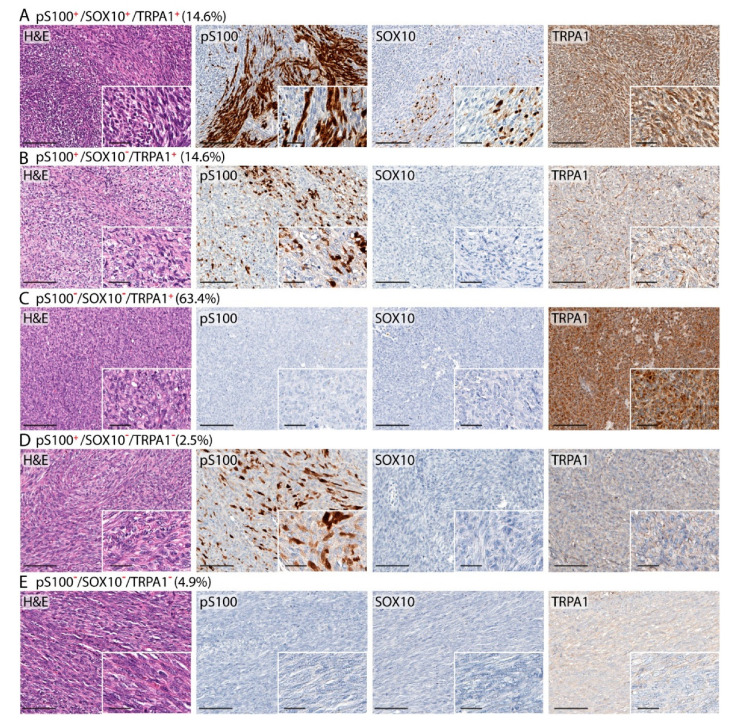
Representative photomicrographs reporting hematoxylin and eosin (H&E) and percentage of SS samples with TRPA1 staining positive to pS100 and SOX10 (**A**); positive to pS100 and negative to SOX10 (**B**); negative to both pS100 and SOX10 (**C**). Representative photomicrographs of SSs negative with TRPA1 staining and positive with pS100 (**D**) or negative with both pS100 and SOX10 (**E**). Magnification ×200, inset ×400 (Scale bars: 100 µm, inset 20 µm).

**Figure 3 biomolecules-10-01446-f003:**
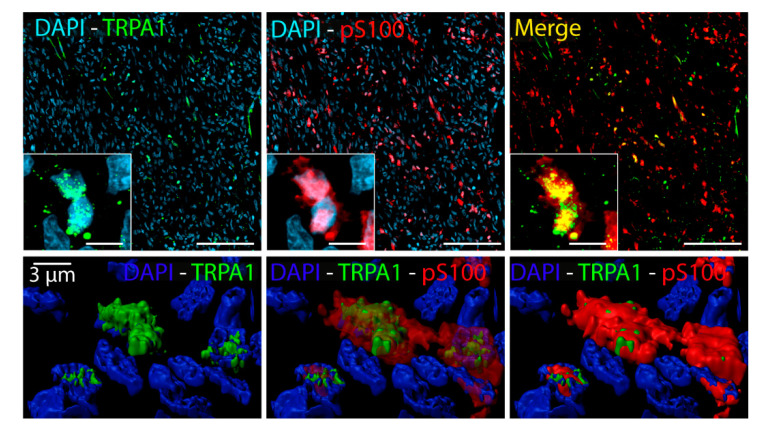
Representative images of a double immunofluorescence staining of pS100 (red) and TRPA1 (green) (Scale bars: 50 µm, inset 10 µm). Bottom panel: representative 3D surface reconstruction of TRPA1 and pS100 positive cells generated from structured illumination microscopic imaging using Imaris Software.

**Figure 4 biomolecules-10-01446-f004:**
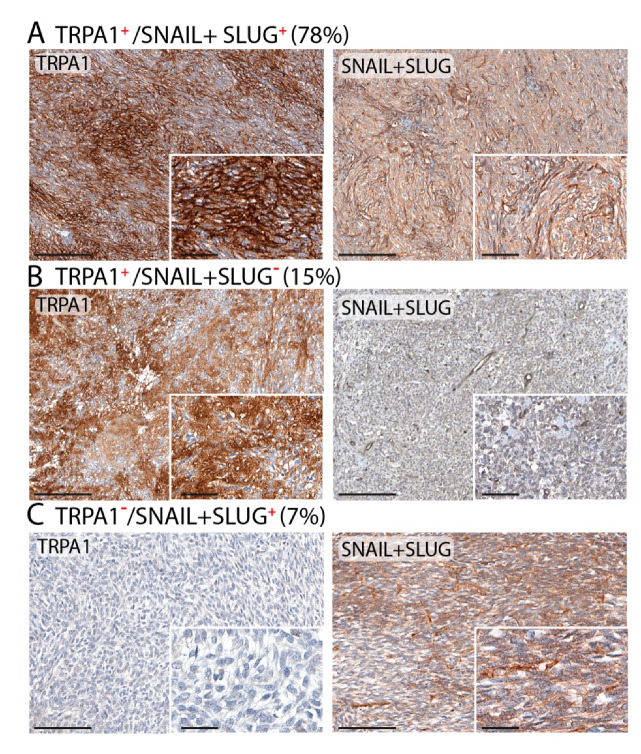
Representative photomicrographs and percentage of SS samples with TRPA1 staining positive with SNAIL+SLUG (**A**), and negative with SNAIL+SLUG (**B**); and negative with TRPA1 and positive with SNAIL+SLUG staining (**C**). Magnification ×200, inset ×400 (Scale bars: 100 µm, inset 20 µm).

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
