# Peer review of "TRPA1 Expression in Synovial Sarcoma May Support Neural Origin"

_biomolecules, 2020, doi:10.3390/biom10101446_

Round 1
Reviewer 1 Report
The revised manuscript by De Logu et al. has provided additional data and sufficient clarity in methodology to address the concerns raised in the previous review.
Reviewer 2 Report
In the revised version of the manuscript “TRPA1 expression in synovial sarcoma may support neural origin”. Authors have quantified the expression of TRPA1 in synovial sarcoma tissues and show a 93% of expression in neoplastic cells. Furthermore, they have incorporated immunohistochemistry samples using specifics neural crest marker antibodies to support that around 78% of synovial sarcoma samples may have a neuronal origin. These results must be taken as suggestive as the neuronal crest marker antibodies have been used together and we do not have them separately. Nonetheless, I believe that the manuscript has improved and provides useful information and thus may warrant publication.
This manuscript is a resubmission of an earlier submission. The following is a list of the peer review reports and author responses from that submission.
Round 1
Reviewer 1 Report
In the manuscript titled “TRPA1 expression in synovial sarcoma may support neural origin”, De Logu et al. describe the immunohistochemical findings of a cohort of human synovial sarcoma biopsies. The authors used antibodies against TRPA1, SOX10, and S100 and found that 14.6% of the samples were stained positive for all three. Only a single antibody for each marker was used, without cross-verification with alternative sources. It should be noted that neo-antigens commonly found in cancerous cells may also lead to false-positive immune-staining results. An unknown percentage of cells expressed both S100 and TRPA1, suggesting that these cells may be of neural crest stem cells, which typically express a multitude of markers, such as neurogenins, FoxD3, Slug, Snail, Notch, etc. Percentages of neoplastic cells versus normal cells in the cohort are also undefined. Whilst most of the samples (63.4%) were stained negative for S100 and SOX10, the relationship between the highly variable TRPA1 expression levels and SS parameters, such as size, malignancy, and prognosis, is also not clear. The analyses were largely qualitative and description, lacking statistical evidence. For instance, what are the statistics and analysis that led to the conclusion that “no correlation between TRPA1 expression and primary vs. recurrent/metastatic nature” (Line 140)? One can only conclude the percentage of cells that express a certain protein using immunohistochemistry data (Figure 1&2). Multiplex immunostaining (Figure 3) is much more informative and quantitative in nature.
Other comments:
- Please indicate the catalogue number or the RRID number of the TRPA1 antibody used in the study.
- Scale bars should be used in the figures.
Reviewer 2 Report
This study attempts to relate the expression of TRPA1 en synovial sarcoma with a neuronal origin. The authors use an histochemistry-based approach to support their hypothesis. The study uses human samples to evaluate the co-localization of TRPA1 immunoreactivity with that of neural markers to conclude that TRPA1 plays a role in pain symptoms and tumor progression. Overall, this is an interesting study that need to address some concerns before warranting publication in Biomolecules.
- Although the overall strategy appears adequate for addressing the question, it appears that the neuronal markers used have not sufficient specificity to unequivocally support the hypothesis and conclusions of the study. For instance, TRPA1 is not a good neuronal marker as it is expressed in many different cell types. Indeed, it is also present in neurons, but other cell types (fibroblasts, etc) also expressed this channel.
- Furthermore, pS199 and SOX10 are rather markers of Schwann cells, which also limits the impact of the results. It would be more appropriate to use more specific neuronal markers to strengthen their results.
- In addition, quantitative account of the images for co-localization should also improve the quality and relevance of the study.
- Expression of TRPA1 as a function of cancer stage would be also important to associate the expression of the channel with tumor progression.